# The Negative Effect of Siblings on Perspective-Taking in Adulthood under Chinese Culture

**DOI:** 10.3390/bs14070599

**Published:** 2024-07-14

**Authors:** Xingyu Fan, Yi Liu

**Affiliations:** 1School of Psychology, Northeast Normal University, Changchun 130024, China; xingyufan678@gmail.com; 2Jilin Provincial Key Laboratory of Cognitive Neuroscience and Brain Development, Changchun 130024, China

**Keywords:** perspective-taking, theory of mind, sibling, parental care, sibling intimacy

## Abstract

Evidence from Western developed countries has consistently found that children with sibling(s) showed better perspective-taking (PT) than only children. However, this was not the case in developing countries like China. Our study investigates whether the potentially hindering effect of having sibling(s) on PT persists into adulthood within the context of contemporary Chinese culture. We employed self-report questionnaires to measure PT, perceived parental care, and sibling relationships among Chinese young adults (21.93 ± 2.35 years old). Our findings indicate that in China, (1) having sibling(s) hinders individuals’ PT in adulthood, (2) a potential mechanism for this effect is that having sibling(s) leads individuals to perceive less parental care during early life, and (3) for those with sibling(s), higher sibling intimacy can enhance PT, particularly among older sister–younger brother pairs. These results suggest that in developing countries such as China, while sibling(s) may diminish PT to some extent, factors like parental care and sibling intimacy can serve as protective factors that mitigate the negative impacts of sibling(s) on PT.

## 1. Introduction

Perspective-taking (PT) is a capacity to consider the world from others’ viewpoints and allows an individual to anticipate the behavior and reactions of others [1]. Through PT, individuals put themselves in others’ shoes to understand others’ cognitive and affective states [2], which may foster successful social interactions and prosocial behaviors [3]. In studies of children’s social development, the false belief task is widely used to measure children’s theory of mind (ToM), which is closely connected with PT [4,5].

Sibling relationships are considered a significant factor influencing the development of ToM or PT. For instance, studies in developed countries like Canada and Britain have shown that children with sibling(s) outperform only children in ToM tasks [6,7]. This advantage persists into adulthood, manifesting as enhanced PT performance for individuals with sibling(s) [8]. However, it is noticeable that the effect of sibling(s) on individuals’ PT was modulated by the families’ socioeconomic status; that is, the positive effect of sibling(s) on PT was particularly evident in families with a high socioeconomic status [9,10,11,12,13]. Thus, it is not surprising that in developing countries such as Iran and China, some studies have not reported positive effects of having sibling(s) on children’s ToM [14,15,16], and others have even noted a negative effect [17]. One possible reason for this downside of sibling(s) in developing countries is that the dilution of resources results in only children receiving more economic and psychological support from their parents [18], which benefits their PT. At present, exploring whether this apparent negative sibling effect in developing countries persists into adulthood is a pressing question. Additionally, examining the roles of parental care and sibling relationships on the effect of sibling(s) on PT is worthy of investigation.

## 2. Literature Review

### 2.1. Presence of Sibling(s) and Perspective-Taking in Developed Countries

The following seminal work provides initial support for sibling effects on children’s ToM. That is, cooperative interactions with siblings could predict children’s ToM performance [19]. Over the past three decades, numerous empirical studies have documented the positive effects of sibling(s) on children’s ToM [6,7,20], with effect sizes ranging from 0.10 to 0.52 [10]. Moreover, studies have investigated whether these sibling effects on PT persist into adulthood. For example, Lo and Mar [8] employed the “Reading the Mind in the Eyes” task to measure PT and found that individuals with sibling(s) (mean age = 24.12 ± 9.68 years) showed better PT performance than individuals without sibling(s), especially for male participants.

It can be seen that Western research largely supports the role of sibling(s) in promoting the development of ToM or PT [6,7,19,20]. Furthermore, this positive effect may extend into adulthood [8]. These positive effects are explained by the increased social interactions that children with sibling(s) experienced compared to only children. Through repeated daily experiences, children gradually acquire the ability to understand other people’s emotions, thoughts, and motivations [21,22]. It is worth noting that these findings are predominantly from developed countries, such as Canada [10] and Britain [7]. It has been shown that the positive effect of sibling(s) is particularly evident in families with a high socioeconomic status [9,10,11,12,13]. Thus, it is necessary to investigate the effects of sibling(s) in developing countries, where living standards are generally lower than in developed countries.

### 2.2. Presence of Sibling(s) and Perspective-Taking in Developing Countries

In developing countries, the situation appears to be different. In China, due to the “Family Planning” policy, research on the impact of sibling relationships has been relatively limited. Existing studies on the association between sibling(s) and PT in China have provided inconsistent evidence. For instance, Zhai and colleagues [17] found that Chinese preschoolers (mean age = 2.61 years) benefited more in PT development from being an only child than from having sibling(s). Hou et al. [23] assessed the ToM performance of preschoolers (mean age = 4.30 years) and found no differences in ToM scores between only children and those with sibling(s). More recently, Zhu et al. [16] observed that the positive effects of sibling(s) on Chinese preschoolers’ ToM were only present in children with low peer interactions. Similarly, in Iran, preschoolers (mean age = 4.79 years) with one or more siblings did not outperform their peers in ToM tasks [15].

These findings suggest that the presence of sibling(s) might not promote, and could even inhibit, the development of PT, contrasting with results from developed countries. It is important to note that these potentially negative effects of sibling(s) on PT have so far only been demonstrated in preschool children. In developing countries, such as China, whether the negative effect of having sibling(s) on PT could extend to adulthood remains unexplored. Furthermore, the mechanisms underlying these negative effects warrant further investigation. 

### 2.3. Parenting for Sibling(s), Sibling Relationships, and Perspective-Taking

A fundamental distinction between families with an only child and those with multiple children lies in the distribution of limited parental time and resources, particularly in developing countries. Evidence suggests that only children experience a unique family upbringing and parenting style distinct from those in multi-child families. Specifically, only children often perceive significantly higher parental emotional warmth compared to their counterparts from multi-child families [24,25]. Additionally, neglect concerning children’s physical, emotional, educational, safety, and medical needs is generally higher in families with multiple children than in those with only one child [26]. These observations indicate that only children receive more focused care from their parents, while individuals from multi-child families often experience a division of parental care among siblings. There is mounting evidence that perceived parental emotional warmth is positively correlated with PT in adolescents [27,28,29] and adults [30]. This raises a crucial question: could the presence of sibling(s) influence parental care, which in turn affects PT later in life? We explore whether parental care might play a mediating role in the association between the presence of sibling(s) and their PT ability.

Although the presence of sibling(s) has demonstrated seemingly negative effects on PT in developing countries like China, sibling intimacy has consistently shown a robust positive impact on PT across various cultures. Specifically, the PT scores of children with sibling(s) are positively correlated with their sibling intimacy; this correlation has been observed in preschoolers [31], primary school students in higher grades [32], and also in middle and high school students [33]. Importantly, similar effects have been noted in adults in Western countries [34]. However, evidence is still lacking in China regarding whether sibling relationships in early life have long-term effects on PT in adulthood.

## 3. Aim of the Study

Based on these findings, the aim of our study is to investigate the impact of having sibling(s) during early years on individuals’ PT in adulthood within the context of contemporary Chinese culture. Specifically, we address three research questions: (1) whether having sibling(s) in early years significantly hinders the level of PT in adulthood, (2) the mediating role of parental care, and (3) whether individuals with sibling(s) benefit in terms of PT from high sibling intimacy. Given that in Western developed countries, the positive effects of having sibling(s) in early life can extend into adulthood [8], we are open to exploring whether the negative effects observed in China might also persist into adulthood. Considering resource dilution, we hypothesized that the reduced parental care for non-only children might mediate the negative effect of sibling(s) on individuals’ PT. Additionally, it is widely established that individuals’ PT can benefit from positive sibling relationships [31,32,33,34]. Therefore, we also hypothesized that, although sibling(s) may have a negative effect on PT in China, positive sibling relationships could serve as a protective factor for the PT of non-only children. To investigate these hypotheses, we conducted a cross-sectional study involving 513 Chinese university students. Their PT, perceived parental care, and sibling relationships were measured through self-reported questionnaires. 

## 4. Method

### 4.1. Participants

A total of 668 young adults from various levels of universities (including public and private, undergraduate, and vocational colleges) across China were initially recruited online for the questionnaires. Of these, 155 (23.2%) participants were excluded due to non-serious responses, identified by repeated responses from the same IP address, excessively short response times (i.e., less than 3 min for all items for participants with siblings; less than 1.5 min for all items for participants without siblings who did not fill in the questionnaires about sibling relationships), and response variances near or equal to zero (less than 0.3 for all items). Ultimately, 513 valid questionnaires were collected (see Table 1 for details), with participants having a mean age of 21.93 ± 2.35 years, and a mean age difference between siblings of 5.73 ± 3.76 years. Participants were compensated between 1.8 and 2.2 yuan. This research received approval from the ethics committee of Northeast Normal University. All data were collected in the year 2023.

### 4.2. Measures

IRI-PT: The PT subscale of the Interpersonal Reactivity Index (IRI) [35] was employed to assess the subjective PT of participants. It consists of 7 items (e.g., “I sometimes try to understand my friends better by imagining how things look from their perspective.”) rated on a 5-point scale ranging from 1 (“completely disagree”) to 5 (“completely agree”). This subscale has been widely used and demonstrated acceptable reliability and validity among Chinese adults [36]. In the current study, the Cronbach’s α coefficient was 0.64. 

PBI-PC: The parental care subscale of the Parental Bonding Instrument (PBI) [37] was used to measure the perceived parental care of participants. This subscale measures the parental rearing styles experienced by individuals before the age of 16. The modified Chinese version of the subscale includes 12 items rated on a 4-point scale from 0 (“completely unlike”) to 3 (“very much like”). This version has been validated with appropriate reliability and validity and is a reliable tool for assessing parenting styles [38]. In the current study, its Cronbach’s α coefficient was 0.87.

ASRQ: The Adult Sibling Relationship Questionnaire (ASRQ) was used to assess individuals’ perceptions of sibling relationships [39]. Participants were asked to recall their upbringing with sibling(s) and to respond to the questionnaire for the sibling closest in age. The questionnaire includes 43 items, divided into three dimensions: sibling intimacy, sibling conflict, and sibling competition. Participants rated how characteristic each item was of their relationship with their sibling on a 5-point scale ranging from 1 (“hardly at all”) to 5 (“extremely”). This scale has been validated for good reliability and validity among Chinese university students [40]. In this study, the Cronbach’s α coefficient was 0.93.

### 4.3. Data Analysis

First, to examine the impact of having sibling(s) on PT, we conducted an independent samples *t*-test to compare PT scores between individuals with and without sibling(s). Second, a similar independent samples *t*-test was conducted to compare perceived parental care between individuals with and without sibling(s). The association between parental care and PT was also tested using Pearson correlation analyses. Third, to further investigate whether the presence of sibling(s) decreases PT through perceived lower parental care, we performed a mediation analysis with the presence of sibling(s) as the independent variable, the PT scores as the dependent variable, and the perceived parental care as the mediator. The mediation analysis was conducted using Model 4 of the PROCESS macro in SPSS 25, with a bootstrap sample size of 5000. Finally, for individuals with sibling(s), we explored whether early sibling relationships were associated with PT levels in adulthood using Pearson correlation analyses. Additionally, we tested whether sibling intimacy could explain the lower PT observed in the older sister and younger brother combination relative to other sibling combinations using another mediation analysis.

## 5. Results

### 5.1. Effect of Presence of Sibling(s)

To examine the impact of the presence of sibling(s) on PT, we compared PT scores between individuals with and without sibling(s). Results indicated that the PT scores for only children were significantly higher than those for individuals with sibling(s) (*t*_(511)_ = 2.653, *p* = 0.008) (Figure 1a). This finding suggests that only children exhibit better PT in adulthood than non-only children in China, which contrasts with findings from Western developed countries [6,7,10,20].

Additionally, the perceived parental care differed between individuals with and without sibling(s). Only children reported significantly higher levels of parental care compared to individuals with sibling(s) (*t*_(511)_ = 2.186, *p* = 0.029) (Figure 1b). Moreover, individuals’ perceived parental care was positively correlated with PT (only children: *r*_(227)_ = 0.243, *p* < 0.001; non-only children: *r*_(282)_ = 0.293, *p* < 0.001, Figure 1c), suggesting that parental care served as a positive factor that promotes PT.

### 5.2. Mediation Role of Parental Care

To further examine whether the presence of sibling(s) decreases PT through perceived lower parental care, we conducted a mediation analysis. The results revealed that the effect of the presence of sibling(s) on parental care was significant (Effect value = −1.1699, LLCI = −2.2214, ULCI = −0.1184), as was the effect of parental care on PT (Effect value = 0.1743, LLCI = 0.1201, ULCI = 0.2286). Importantly, a significant direct impact of sibling presence on PT was observed (Direct effect = −0.7155, LLCI = −1.3751, ULCI = −0.0559), and the indirect effect of sibling presence on PT through parental care was also significant (Effect value = −0.2040, SE = 0.1019, LLCI = −0.4254, ULCI = −0.0238) (Figure 1d). These findings suggest a partial mediating effect, indicating that non-only children in China exhibited lower PT in adulthood, partly because they perceived less parental care.

### 5.3. Effect of Sibling Intimacy

For individuals with sibling(s), we explored whether early sibling relationships were associated with PT levels in adulthood. Our results demonstrated a positive correlation between PT levels and sibling intimacy (*r*_(282)_ = 0.254, *p* < 0.001) (Figure 2a), and a negative correlation with sibling conflict (*r*_(282)_ = −0.210, *p* < 0.001). However, sibling competition showed no significant correlation with PT (*r*_(282)_ = −0.005, *p* = 0.939). These findings suggest that positive sibling relationships in early years can promote PT in adulthood.

Interestingly, our dataset showed that the proportion of elder sister–younger brother combination (43.0%) in two-child families was relatively higher than other combinations (elder brother–younger brother: 19.3%, elder brother–younger sister: 18.3%, and elder sister–younger sister: 19.3%, see Table 1). Moreover, the elder sister–younger brother group also exhibited lower sibling intimacy (*t*_(200)_ = 2.914, *p* = 0.004; Figure 2b) and lower PT (*t*_(200)_ = 2.020, *p* = 0.045; Figure 2c) than other combination groups. Furthermore, we found that sibling intimacy mediated the effect of elder sister–younger brother combination (vs. other combinations) on PT. Specifically, the effect value of combinations on sibling intimacy was significant (Effect value = 7.4948, LLCI = 2.4223, ULCI = 12.5672), as was the effect of sibling intimacy on PT (Effect value = 0.0485, LLCI = 0.0213, ULCI = 0.0757). The direct impact of sibling combinations on PT was not significant (Effect value = 0.6790, SE = 0.5126, LLCI = −0.3318, ULCI = 1.6898), while the indirect impact through sibling intimacy was significant (Effect value = 0.3636, SE = 0.1728, LLCI = 0.0940, ULCI = 0.7590) (Figure 2d). These results indicate a complete mediating effect, showing that the elder sister–younger brother combination experienced lower PT due to reduced levels of sibling intimacy.

As the sibling conflict between the elder sister–younger brother group and other combinations was not significant (*t*_(200)_ = 0.550, *p* = 0.583), we did not analyze the mediation role of sibling conflict.

## 6. Discussion

This study explored the association between having sibling(s) and individuals’ PT in adulthood within the Chinese cultural context. Our findings reveal the following: (1) In China, having sibling(s) hinders PT in adulthood, (2) a potential mechanism is that having sibling(s) leads individuals to perceive less parental care in early life, and (3) for individuals with sibling(s), higher sibling intimacy is associated with higher PT in adulthood. Specifically, among older sister–younger brother pairs, lower sibling intimacy may account for their reduced PT in adulthood. These results indicate that in China, although having sibling(s) may decrease PT to some extent, both parental care and sibling intimacy serve as protective factors that can mitigate the negative impact of sibling(s) on PT. 

In developed countries, it has been demonstrated that individuals with sibling(s) tend to have higher PT abilities [6,7,10,20]. However, Zhai and colleagues [17] demonstrated that in China, having sibling(s) hindered the development of PT for preschoolers. Complementing their results, our study is the first to show that the negative effect of having sibling(s) on PT could extend into adulthood, with a potential mechanism being the decreased perceived parental care among non-only children. It is well known that in collectivistic culture, interpersonal relationships are prioritized over individual uniqueness [41] and research suggests that individuals from collectivistic cultures may be less egocentric and better at taking others’ perspectives than those from individualistic cultures [42,43]. Regarding the collectivistic culture of China, sibling relationships are considered crucial within the family system and closely intertwined with carefully defined roles, as reflected in the idiom “an elder brother is like a father” [44]. Despite these factors, why do siblings in China still exhibit a negative effect on individuals’ PT? Perhaps considering the richness of resources is necessary. Notably, our participants were born at the turn of the 21st century. During this period, despite rapid economic development, many Chinese families experienced relatively low living conditions. Families with more than one child may have faced increased financial burdens, leading to parents being preoccupied with work and unable to adequately attend to their children’s needs. This parenting stress may negatively predict children’s social skills [45]. There is also mounting evidence that a family’s socioeconomic status can modulate the association between siblings and ToM, with the positive effects of having sibling(s) being particularly evident in families with a high socioeconomic status [9,10,11,12,13]. According to the Resource Dilution Theory [18], only children are likely to receive more resources—not just economic, but also more timely responses to their needs, greater emotional care, more patience, and more psychological support. Such comprehensive parental care could enhance the development of PT in only children. Conversely, children with sibling(s) may often compete for limited parental resources, leading to a more self-centered mindset and lower PT in adulthood. Therefore, it is plausible that the reduced parental care experienced by children with sibling(s) could hinder their PT development in China and other developing countries. 

Although individuals with sibling(s) generally exhibit lower PT compared to only children, there is a beneficial aspect: positive sibling relationships, such as sibling intimacy, can promote PT. This aligns with findings from studies in developed countries [34]. It suggests that fostering a harmonious family environment and enhancing sibling intimacy are crucial for improving PT among children with sibling(s). High-quality sibling relationships often encourage the adoption of constructive strategies like negotiation and reduce the reliance on destructive tactics such as aggression to resolve conflicts [46]. Constructive conflict resolution requires siblings to consider each other’s feelings and express their viewpoints, which can facilitate the development of PT [46,47,48]. It is noteworthy that our study revealed a high prevalence of the older sister–younger brother combination, which exhibited lower sibling intimacy and lower PT compared to other sibling combinations. This suggests that parents are more likely to have a second child when the first is a girl. Furthermore, the low sibling intimacy observed between older sisters and younger brothers appears to be a significant factor contributing to their reduced PT in adulthood. One possible reason for this prevalence is that girls are generally considered more obedient and well-behaved than boys, potentially making parents feel more confident about having another child. Alternatively, in China, some parents or grandparents place higher value on male offspring. Thus, if their first child is a girl, they are more inclined to have another child, hoping for a boy. The lower sibling intimacy between older sisters and younger brothers may stem from unequal parental treatment, stereotypical roles (e.g., older sisters being encouraged to modestly support their younger brothers), or gender differences in interaction (e.g., sisters not sharing their secrets with their brothers). These findings underscore the importance of enhancing sibling intimacy, particularly between older sisters and younger brothers, as it is crucial for their PT in adulthood. Given the high prevalence of the older sister–younger brother combination, improving intimacy between them might effectively mitigate the negative impact of sibling(s) on individuals’ PT.

In addition, it is particularly important to emphasize that our participants are young adults. Typically, siblings spend a significant amount of time together during their early years. However, as they enter early adulthood, they may grow apart, focusing more on their personal relationships, careers, and establishing their own families. This may explain why the correlation between sibling intimacy and PT is significant but not strong. Despite this, there is evidence suggesting that the shared experiences and histories formed in early sibling relationships continue to profoundly influence their relationship quality and subtly impact their psychological and behavioral outcomes in adulthood [49,50]. Our study is the first to demonstrate that the negative effects of having sibling(s) on PT can persist into adulthood in China, underscoring the long-lasting effects of early sibling relationships on future social skills.

Taken together, our findings have significant implications for practice. First, we remind parents of the potential long-lasting negative impacts of having a second child, especially for families in developing countries. For parents who already have one child, it is crucial to carefully consider whether they are prepared to provide ample and equal care and support to each child to mitigate the negative impact of sibling(s) on PT. Second, in multi-child families, parents and other caregivers should focus on fostering sibling intimacy. This can be achieved by increasing contact and companionship between siblings, creating a family environment conducive to communication and dialogue, guiding children in playing games they enjoy together, encouraging their joint participation in various family activities, and promoting cooperation and mutual assistance, especially in families with an elder sister and a younger brother. For individuals with sibling(s), it is important to recognize that establishing high intimacy with siblings can benefit one’s PT in adulthood, which is also crucial for other interpersonal relationships in social life.

Finally, it is crucial to acknowledge some limitations and directions for future research. Firstly, our study exclusively tested Chinese young adults who were all university students. Whether the observed negative effects of sibling(s) on PT are also present in non-university students in China and in other developing countries remains to be explored to provide a more comprehensive understanding of the global implications of sibling dynamics on PT. Secondly, our explanations based on resource dilution theory are speculative. Other unique aspects of Chinese society, such as cultural orientation or parenting strategies, might also influence the results and differentiate them from those observed in Western developed countries. For instance, Chinese culture emphasizes interpersonal harmony and obedience to authority figures, such as parents or older siblings, while discouraging the expression of personal opinions [14,51]. This cultural norm could potentially reduce children’s exposure to and participation in discussions about mental states with family members, including siblings [52,53]. These cultural factors warrant further empirical investigation to understand their impact on the sibling effects on PT. Thirdly, since 2013, China has started to relax its one-child policy [54]. This historical context may limit the generalization of our conclusions. As China progresses through policy reforms and sustained economic growth, improvements in resource availability in family life are expected, which could enhance the parenting environment in families with multiple children. Therefore, it is necessary to investigate whether sibling(s) have a similar effect on PT in children born under these improved conditions when they reach adulthood. Fourthly, our study relied solely on the PT subscale of the Interpersonal Reactivity Index [35] to measure subjective levels of PT, which reflects a combination of the tendency and ability of individuals to adopt the perspectives of others. In real-life scenarios, some individuals may possess the ability but not the inclination to adopt others’ perspectives. Differentiating the effects of sibling(s) on the tendency versus the ability to engage in PT could yield deeper insights. Lastly, other potential factors such as sibling gender, the age difference between siblings, and birth order may also influence the effects of sibling(s) on PT. A thorough exploration of these factors could provide a more comprehensive view of the complexities involved in how sibling(s) affect PT.

## Figures and Tables

**Figure 1 behavsci-14-00599-f001:**
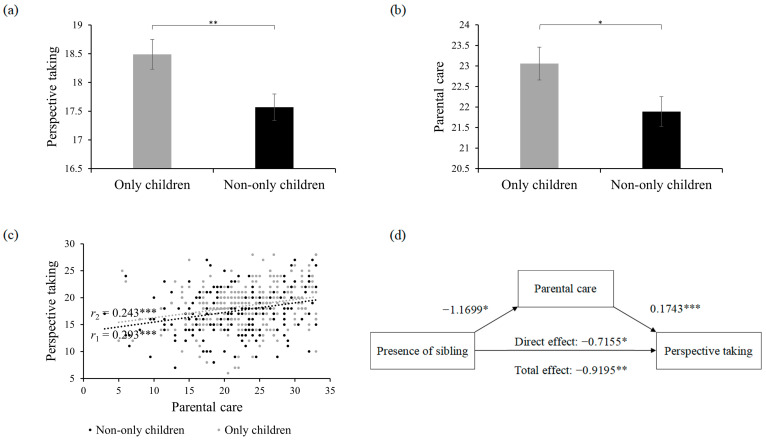
(**a**) Perspective-taking scores for only children and non-only children groups. (**b**) Perceived parental care for only children and non-only children groups. (**c**) Correlations between perspective-taking and parental care for only children (gray dots) and non-only children (black dots). (**d**) Mediation role of parental care on the association between presence of sibling(s) and perspective-taking. The error bars represent standard errors. * *p* < 0.05, ** *p* < 0.01, *** *p* < 0.001.

**Figure 2 behavsci-14-00599-f002:**
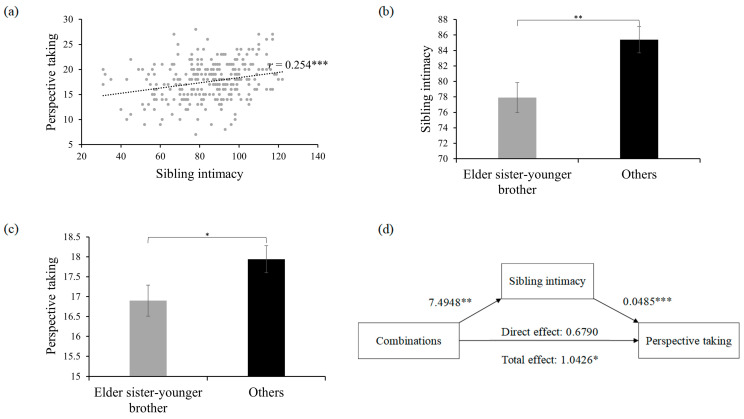
(**a**) Correlation between perspective-taking and sibling intimacy. (**b**) Sibling intimacy for the elder sister–younger brother group and other combinations. (**c**) Perspective-taking for elder sister–younger brother group and other combinations. (**d**) Mediation role of sibling intimacy on the association between sibling combinations (i.e., elder sister–younger brother combination vs. other combinations) and perspective-taking. The error bars represent standard errors. * *p* < 0.05, ** *p* < 0.01, *** *p* < 0.001.

**Table 1 behavsci-14-00599-t001:** Information of participants.

Participant Characteristics	Levels	N	Percentage
Gender	Male	246	48%
	Female	267	52%
Region	North	325	63.4%
	South	188	36.6%
Sibling status	Have sibling(s)	284	55.4%
	None	229	44.6%
Number of siblings (For participants having siblings)	1	202	71.1%
	More than 1	82	28.9%
Combinations (in two-child families)	An elder brother and a younger brother	39	19.3%
	An elder sister and a younger brother	87	43.1%
	An elder brother and a younger sister	37	18.3%
	An elder sister and a younger sister	39	19.3%

## Data Availability

Data are available at https://github.com/Xingyu-Fan/Data-of-Sibling-Research (accessed on 8 June 2024).

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
