# Peer review of "The Negative Effect of Siblings on Perspective-Taking in Adulthood under Chinese Culture"

_behavsci, 2024, doi:10.3390/bs14070599_

Round 1

Reviewer 1 Report

Comments and Suggestions for Authors

Thank you for the opportunity to read this research manuscript titled “Siblings Influence Perspective-Taking in Adulthood under Chinese Culture.” This manuscript shows the hindering effect of having siblings on perspective taking in adulthood in Chinese culture. I commend the authors for exploring the understudied area of sibling relationships during the important life stage of adulthood. This is an especially interesting area of study as having a relationship with siblings in adulthood is based much more on individual choice rather than other environmental pressures that may have existed in previous life stages. It is an important study, but suffers from some flaws, as shown below:

1.       I recommend the authors include a theory that structures your work and highlights why you chose to examine these variables. For instance, in the discussion section, the authors mention Resource Dilution Theory first time but there is nothing about it in the introduction.

2.      Also, it would be more reader-friendly if the authors provided detailed information about the definition of perspective taking, Theory of mind, or maybe cognitive empathy. Go into more detail on how why understanding PT is important?

3.      Page 2, line 47-49, … "Reading the Mind in the Eyes" task to measure PT and found 47 that individuals with siblings show better PT performance than individuals without siblings, especially for male participants.’’ Please mention the age range of participants in this study.

4.      The authors claim that Eastern culture is different than Western culture in terms of the role of sibling relationships in PT. There seems to be a lack of cross-cultural studies in this area, but please provide examples of cross-cultural studies regarding sibling relationships and regarding PT, separately.

5.      Please provide hypotheses for the research questions. It is surprising that the authors did not share the mean age of the participants who completed the questionnaires (and mean age gap between siblings if known). The authors mentioned that the participants were Chinese university students. So, instead of adulthood which is a very large of life, the authors may refer to participants as young adults, depending on the age of the participants. Additionally, did the authors consider or actually recruit young adults who were not in college?

6.      Did participants receive any incentive?

7.      What were the inclusion and exclusion criteria?

8.      Were there any differences between the participants included in the study and those not included in the study (e.g., did not reply, screened out by authors)?

9.      What type of university (public/private, large/small, community, etc.)?

10.  What year was the data collected?

11.  Please provide information about the reliability and validity of the study questionnaires. Were the multiple tests accounted for and corrected for by using a Bonferroni correction or something of the sort?

12.  Some correlations for coefficients for predictor and outcome variables are significant, but not strong. Please discuss it.

13.  The authors show that higher sibling intimacy could enhance PT, particularly among older sister-younger brother pairs. An elder sister and a younger brother pair is almost half of the participants who have siblings. Please discuss the role of this on the results in more detail although it was mentioned in the discussion in a sentence.

14.  Page 6, line 200, the authors mentioned that this study explored how sibling presence influences PT in adulthood within the Chinese cultural context. Please avoid using the term influence throughout the manuscript since it is not a longitudinal study.

15.  Page 6, line 214, the authors highlighted that family’s socioeconomic status can modulate the association between siblings and ToM, with the positive effects of siblings being particularly evident in families with high socioeconomic status. If this is the case, please also provide information about SES in the introduction as well.

16.  Further, the current manuscript has weak implications for practice and little to no description on why studying this topic is important to sibling relationships in adulthood, and thus I do feel that it would be beneficial to strengthen these areas to highlight the contribution of this study into the sibling field.

Comments on the Quality of English Language

The authors should review standard formatting and citation guidelines recommended by the journal, as well as standard grammar and punctuation (i.e., colons in place of semicolons, misplaced commas, etc.) throughout the manuscript. 

Reviewer 2 Report

Comments and Suggestions for Authors

Thank you for the opportunity to review your manuscript. The topic is compelling and important. 

I found the writing of the manuscript to be very clear, succinct, and easy to follow.

The literature review was clear and directly tied to the subsequent study aims.

For the method the variables are clearly identified.

it might be helpful to write up an analytic plan paragraph as opposed to have the analytic method in each section alone. Because the figures some before the plan in some instances (mediation) I wanted to know what mediation method was used but it did not come up until later. 

The results were clear and easy to follow.

The discussion was also well written and helped to explain some of the findings further. I especially appreciated the connection to cultural differences around obedience and opinion sharing, and whether that might play a role in this type of work. 

Author Response

Comments: It might be helpful to write up an analytic plan paragraph as opposed to have the analytic method in each section alone. Because the figures some before the plan in some instances (mediation) I wanted to know what mediation method was used but it did not come up until later. 

Responses: We are truly grateful for the high evaluation of our work! We have added a paragraph on the analytic plan in the methods section (line 166-180).